# Effects of Accelerated Aging on Color Stability and Surface Roughness of a Biomimetic Composite: An In Vitro Study

**DOI:** 10.3390/biomimetics7040158

**Published:** 2022-10-09

**Authors:** Abdullah Alshehri, Feras Alhalabi, Mohammed Mustafa, Mohamed M. Awad, Mohammed Alqhtani, Mohammed Almutairi, Faisal Alhijab, Carlos A. Jurado, Nicholas G. Fischer, Hamid Nurrohman, Abdulrahman Alshabib

**Affiliations:** 1Department of Conservative Dental Sciences, College of Dentistry, Prince Sattam Bin Abdulaziz University, Al-Kharj 11942, Saudi Arabia; 2Dental Intern, College of Dentistry, Prince Sattam Bin Abdulaziz University, Al-Kharj 11942, Saudi Arabia; 3Woody L. Hunt School of Dental Medicine, Texas Tech University Health Sciences Center, El Paso, TX 79905, USA; 4Minnesota Dental Research Center for Biomaterials and Biomechanics, University of Minnesota School of Dentistry, Minneapolis, MN 55455, USA; 5Restorative Dentistry, A. T. Still University Missouri School of Dentistry & Oral Health, Kirskville, MO 63501, USA; 6Department of Restorative Dentistry, King Saud University College of Dentistry, Riyadh 11545, Saudi Arabia; 7Engr Abdullah Bugshan Research Chair for Dental and Oral Rehabilitation, King Saud University, Riyadh 11545, Saudi Arabia

**Keywords:** blending, color, surface roughness, resin composite, aging

## Abstract

The aim of this in vitro study is to compare the color stability and surface roughness of conventional and self-blending resin composites before and after staining and aging. Three conventional composites (Filtek Z350, IPS Empress Direct, and Estalite Palfique LX5) and one self-blending (Omnichroma) resin composite were used in this study. Sixty discs were prepared and polymerized in a metal mold (*n* = 15 per group). Samples were then finished and polished by Layan discs. Color testing and roughness testing were measured as a baseline (T0) by a spectrophotometer and profilometry. Samples were then stained with tea for 24 h, water aged for 30 days, and then a second reading (T1) was performed. Finishing and polishing were performed again, and a third reading (T2) was collected. All groups showed significant decrease in all color parameters (L*, a*, and b*); however, after polishing, all groups showed color enhancements matching pre-experiment baseline colors in all color parameters (L*, a*, and b*), except for Estelite Palfique LX5, which showed a significant difference in L relative to the baseline. Furthermore, Estalite Palfique LX5 showed increased roughness after staining compared to the baseline, unlike other groups. No significant differences in color stability were found between self-blending composites and other composite materials. Accelerated aging and staining had minimal effects on the surface roughness of self-blending composite.

## 1. Introduction

With advancements in dental materials, dental composite popularity and use has grown. Dental composite resins have long dominated restorations because of large improvements in the field of esthetic dentistry both in terms of physicomechanical and esthetic characteristics. The color stability of dental resin composites in a dynamic oral environment is a key criterion affecting their clinical longevity. However, dental composite resin’s extrinsic and/or intrinsic staining is one of their esthetic limitations [1,2,3]. Superficially stained restoration can be solved by polishing to retain the initial matched color; otherwise, a replacement of the restoration may be indicated [4]. Overall, poorly matched color is one indication for dental composite resin replacement [5].

Restoration composition, filler characteristics, oral hygiene, diet, smoking, finishing, and polishing determine the surface roughness of composite resin [6,7]. A surface roughness that is higher than 0.2 µm is considered a bacterial plaque retentive area [8]. Rough restoration surfaces can improve the adhesion of bacteria, increase rates of gingivitis, and cause secondary caries [9,10,11,12]. Moreover, tongue, lips, and cheeks irritation can occur from rough restoration surfaces [13]. Thus, roughness is an overall important feature in the design of dental resin composites.

A major component of resins composites includes fillers. Fillers play a main role in the reduction in shrinkage after polymerization and thermal expansion and plays a crucial role in lowering water absorption and solubility. These fillers are also important for the physical and the aesthetic properties of the composites, in addition to polishing abilities. Improved filler qualities allow the material to withstand changes that are frequent in the oral cavity and, hence, help in the ability to withstand chewing while retaining aesthetics [14]. One relevant filler technology includes self-blending resin composites. These is a single-color blending composite that uses smart chromatic technology to match tooth color by generating structural colors rather than adding color to the material, mimicking the physical qualities of light and not pigments or dyes [15]. 

Resin composites were not favoured over the amalgam when they were initially introduced to the market as they exhibited inferior physical properties. Over time, many modifications have been applied that made the composites more desirable than amalgams, such as single-color blending composites. There are more conventional composites, and there are many variants of the included fillers, such as nanohybrid composites, conventional composites, and micro-hybrid composites [16].

In recent times, these new variants combine the best properties of various other composites; these are termed “nano-filled composites.” These new restorative materials are made of nano-sized particles in the particle size range of 1–100 nanometers that possess many physical and mechanical properties that are essential for withstanding the masticatory forces of the oral cavity, while still maintaining tooth-like appearances. In addition, they are comparatively easier in application than previous technique-sensitive resin composites. One of the best features of these nanomaterials is their greater polishing ability and smooth surface texture. This is due to the reduced interstitial spaces among inorganic particles that are amongst nanoparticles [17]. Filtek Z350 is a direct light-cured nanohybrid composite that is commonly used for the restoration of both posterior and anterior teeth [18]. IPS Empress Direct is a highly aesthetic nanohybrid composite for the restoration of posterior and the anterior teeth [19]. On the other hand, Omnichroma and Estalite Palfique LX5 are both supra-nano spherical filler composites that have been recently introduced, which have been known to produce greater color stability with a single shade. The most important step in the application of the composite restoration is the finishing and polishing phases, which help in improving aesthetics and longevity [20,21]. Indeed, poorly finished surfaces may present accumulation points for plaque and its detrimental effects. 

The aim of this in vitro study is to evaluate and compare color stability and surface roughness between three different nano-filled composite materials and one self-blending resin composite. The investigated null hypothesis is no difference between different dental composite materials in terms of color stability and surface roughness: (1) between composite resin groups and (2) within composite resin groups subjected to stain and aging.

## 2. Materials and Methods

### 2.1. Sample Preparation

Sixty samples shaped into discs (8 mm diameter by 2 mm in depth) were randomly distributed to four groups (*n* = 15 per group): group A (Estelite Palfique LX5, Tokuyama Dental, Tokyo, Japan), group B (Estelite Omnichroma, Tokuyama Dental, Japan), group C (IPS impress Direct, Ivoclar Vivadent, Liechtenstein, Zurich), and group D (Filtek Z350, 3M Oral Care, St. Paul, MN, USA) (details in Table 1).

To standardize the size, samples were prepared in a metal mold (2.0 mm in depth and 8.0 mm in diameter). Light curing was performed following the manufacturer’s instructions in five overlapping points (40 s each) on the top surfaces starting from the periphery and moving sideways to the opposite end of the specimen using an EliparTM S10 (3M Oral Care, St. Paul, MN, USA) at 1000 mW·cm^2^ for 40 s for each sample. Light intensity was assessed with a radiometer (SDS/Kerr, Orange, CA, USA). Samples were finished and polished using Layan discs (Layan, Saudi Arabia) at 20,000 rpm for 10 s for each disc (black coarse, red medium, green fine, and white superfine) to create baseline (T0) samples. Color and roughness tests were conducted at different time points (T0, T1, and T2). 

### 2.2. Staining Process

A tea solution was used by steeping two black-tea bags (2 g each) into 300 mL of boiling water for 5 min. Tea bags were removed, and the solution was left to cool down for 5 min. Samples were immersed in the same tea solution for 24 h.

### 2.3. Aging Process

After the completion of the staining process, samples were rinsed and immersed in a distilled water at room temperature and replaced every 24 h for 30 days. After staining and aging processes were completed, color and roughness tests were conducted for the reading after baseline (T1).

### 2.4. Finishing and Polishing

The samples were again finished and polished by (Layan discs, Layan) by using a low-speed handpiece (20,000 rpm) at 10 s for each disc. The kit has 4 discs (black coarse, red medium, green fine, and white superfine). Samples were rinsed and dried between each disc. Color and roughness tests were conducted again (T2).

### 2.5. Color Testing

The specimen’s color was observed by using a digital spectrophotometer (Easyshade, VITA, USA) under a standardized environment using a color-matching box (GTI MiniMatcherR, GTI Graphic Technology, Inc., Newburgh, NY, USA). The total color change was calculated using the following formula: ΔE = (ΔL*^2^ + Δa*^2^ + Δb*^2^)^1/2^.

### 2.6. Surface Roughness 

All samples were subjected to surface profilometry using a Contour GT-K 3D Optical Microscope (Bruker, MA, USA) with associated Vision 64 software. Samples were measured by vertical scan interferometry using 5× Michelson magnification lens with a field of view of 1 mm × 1 mm, Gaussian regression filter, a scan speed of 1×, and thresholding of 4. Each sample was scanned at 3 equidistant points and averaged accordingly to determine roughness (Ra) values. 

### 2.7. Statistical Analysis

Data were coded and entered using the statistical package for the Social Sciences (SPSS) version 25 (IBM, Armonk, NY, USA). Quantitative variables were summarized by a median as a measure of central tendency and range as a measure of dispersion. According to distribution of variables using the Kolmogorov–Smirnov test, the following tests were used: The Kruskal–Wallis test was usen for the comparison of more than 2 groups in abnormally distributed variables. For significant results, pair-wise comparisons were conducted using the Mann–Whitney U test. The Friedman test was used for abnormally distributed variables that measured more than two times. For significant results, pair-wise comparisons were performed using Wilcoxon test. A level of significance set at 0.05 was used for all statistical tests herein.

## 3. Results

### 3.1. Overall Color Measurement

The purpose for this study was to evaluate the color stability of different dental composites when subjected to staining and water aging (Figure 1). Following tea solution staining and water aging, all groups showed significant decreases in all color parameters (L*, a*, and b*) (*p* < 0.0001); however, after polishing, all groups showed color enhancements matching pre-experiment baseline colors in all color parameters (L*, a*, and b*) (*p* > 0.05), except for group A, which showed significant differences in L relative to the baseline (*p* < 0.0001).

Table 2 shows color changes starting from the baseline color to the color obtained after staining and then after polishing in each group separately, as there is highly statistically significant difference between episodes, such as (*p* < 0.0001), in all parameters (L*, a*, and b*). Pair-wise comparisons were performed using the Wilcoxon test, as shown in detail in Table 1.

The assessment of color change (after staining Vs. after polishing) was analyzed in each group separately (ΔL*, Δa*, Δb*, and ΔE). Statistically significant differences were detected in all color parameters in all groups between the two episodes, and the high value of ΔE in the groups at staining suggests a clinically significant difference from baseline measurements, while low ΔE values after polishing indicate no clinically significant differences from baseline measurements (Table 3).

### 3.2. Surface Roughness Measurement

Table 4 shows comparisons between roughness in different episodes starting from baseline measurements until after polishing in each group separately. There was only a statistically significant difference in group C (X^2^_Friedman_ = 16.8, *p* = 0.0001), by using pairwise comparisons between roughness at baseline vs. roughness after stain.

Table 5 shows comparisons between groups with respect to roughnesses as there is highly statistically significant difference between groups at baseline, after staining, and after polishing as ((H = 24.9, *p* = 0.0001), (H = 25.2, *p* = 0.0001) and (H = 27.5, *p* = 0.0001), respectively) by using pair-wise comparisons. Significant differences detected between {Group C vs. {group A& group B & group D} are shown in Figure 2.

## 4. Discussion

In this in vitro study, three different nano-filled resin composite materials and one self-blending resin composite were evaluated and compared in terms of color stability and surface roughness in three phases: (1) baseline after curing, finishing, and polishing (T0); (2) after staining and aging treatments (T1); and (3) after finishing and polishing (T2). Comparisons were performed between and within all groups of study. The tested hypotheses were as follows: (1) There is no difference between different dental composite materials in terms of color stability, and (2) there is no difference between different dental composite materials in terms of surface roughness. Based on the obtained results, the first hypothesis was accepted because, after polishing, all groups showed color enhancements matching pre-experiment baseline colors. The second hypothesis was rejected as there was a highly statistically significant difference in roughness between groups at baseline, after staining, and after polishing, such as ((H = 24.9, *p* = 0.0001), (H = 25.2, *p* = 0.0001), and (H = 27.5, *p* = 0.0001), respectively). 

When composites lose their mechanical properties, the loss leads to inferior clinical performances as well as shapes and colors. This will in turn lead to poor aesthetics [22]. When chemical properties are lost, the material becomes prone to wear and is easily abraded, thus affecting material longevity. These properties are dependent on various challenges that the material has to endure intraorally. The same conditions may be replicated in vitro; however, this may not be completely identical. In the present study, saliva was simulated with distilled water. The ionic quality of the food, enzymes in saliva, pH, and the heat of the food may all influence the composite’s properties and were not tested here. This is a limitation of the study; hence, the study may not be directly compared to intraoral conditions. 

Color plays a crucial part in achieving optimal aesthetics. The development of restorative materials with highly aesthetic characteristics and their widespread usage in dental practice stemmed from an increase in patients’ desire for aesthetics. However, one of the most significant disadvantages of resin composites is their proclivities for discoloration, which may be a key reason for the need to replace restorations. As a result, restorative materials should closely match the initial hue and maintain the cosmetic resemblance in the repaired tooth throughout time and challenges. The discoloration of composite materials is linked to the composition of resin fillers, resin matrices, and staining agents [23,24,25,26,27,28,29,30,31].

Resin composite materials that can absorb water can also absorb other fluids and fluid-borne agents, causing discoloration. While the resin matrix of composite materials may absorb water from the environment into the majority of their structure, inorganic glass fillers can only absorb water on their surface and cannot absorb water into the bulk of the material. However, by expanding and plasticizing the resin component, hydrolysing the silane, and promoting micro-crack developments, excessive water sorption might shorten the life of a resin composite. As a result, stain penetration and discoloration are allowed through micro-cracks or interfacial gaps at the filler–matrix interface [27].

It has been founded in the literature that Omnichroma has demonstrated significant color parameter changes after storage in different aging and/or staining solutions [32,33,34,35,36]. Omnichroma showed the lowest color change after accelerated aging and great color stability in all color parameters in a previous comparative study [35]. On the other hand, IPS Empress Direct showed increased color change after staining than other resin composites [36] but demonstrated good color stability and staining resistance after clinical aging in another study [37]. Filtek Z350 and IPS Empress Direct showed a significant color change after staining [38]. Finally, Filtek Z350 showed more color parameter differences than that of IPS Empress Direct in a final comparative study [39].

After 30 days of immersion in each of the three types of solutions, the three composite materials in this investigation showed statistically significant color changes. The findings differ from Aldharrab et al.’s [29], who found that the color shift of composite resins submerged in Red Bull was statistically negligible, while Charisma Classic’s hue was the most susceptible to alteration. Tokuyama Omnichroma, on the other hand, showed the least effect from immersion; this could be explained by monomer contents in the mentioned restorative materials, as other resins contain BisGMA, which has higher water sorption than UDMA and TEGDMA, which are monomer ingredients of Estalite Palfique LX5 and Tokuyama Omnichroma, respectively. These effects from monomers are consistent with the results from previous studies [24,25,29,30,31,32,33].

Using Smart Chromatic Technology, Tokuyama Omnichroma achieves a wide range of color-matching capabilities. Tokuyama Omnichroma has consistently sized 260 nm spherical fillers that enable “Smart Chromatic Technology.” Structural color is formed when the structure of a material intensifies or reduces particular wavelengths of light, resulting in colors that are distinct from what the substance is. The highest color change was observed in all samples submerged in coffee, followed by tea and distilled water. This is the case because coffee contains a large amount of artificial coloring. All composites submerged in coloring liquids demonstrated a statistically significant color shift towards the yellow axis with aging time when comparing the Delta b, which shows color shifts between the blue and yellow Axis. Samples submerged in the coffee beverage have the greatest value. The Tokuyama-Omnichroma-tested samples had the lowest value. However, samples immersed in distilled water revealed a statistically significant color shift toward the blue axis, which may be explained by the samples’ water sorption and the absence of any pigmenting substances in distilled water. 

The mechanical and chemical properties of the material influence its surface roughness. The surface was smoother at the baseline for these composite resin materials as compared to specimens immersed in staining media. However, it was only significant for Estalite Palfique LX5. The particle’s size and shape affect the surface roughness of resin composites. This was evident in the present study as a variation for all four groups. It is thought that when softer resin materials are exposed to staining materials that have a lower pH, filler particles are exposed, which will lead to increased surface roughness.

Estalite Palfique LX5 is made of TEGDMA and Bis-GMA. Bis-GMA has lower water solubility and absorption properties, while TEGDMA is a more hydrophilic monomer that shows water absorption. Composites containing TEGDMA show a general relationship wherein the storage modulus decreases with immersion time proportionally to the absorbability of the material. Hydrophilic groups such as the ethoxy group in TEGDMA are thought to show affinity with water molecules by hydrogen bonding [27]. Estalite Palfique LX5 showed the greatest change in surface roughness in tea and distilled water due to its surface hydrophilicity with respect to the TEGDMA monomer, which increases water uptake. The observed results of the present study are in accordance with the observations made in the study of Al Ghamdi et al. [28].

It has been shown in the literature that Omnichroma’s surface roughness does not change after coffee staining [39]. However, it has been founded that Monster energy drinks greatly affect the surface roughness of Omnichroma [33]. Others found no statistical differences in surface roughness for both Omnichroma and Filtek Z350 after coffee staining [40]. Of course, material surface roughness is dependent on the type of material, and Filtek Z350 has been shown to have lower surface roughness than other resin composites [41]. The same results were observed for Filtek Z350 after in vitro aging as well [42]. IPS Empress Direct has been shown to not change surface roughnesses after coffee, tea, and distilled water immersion [35]. This is an in vitro study that is presented with some limitations. More staining products would yield a more comprehensive outcome; the aging method used is valid but future research could test different aging methods and evaluate the effect. Different shades and material brands could also be tested as a continuation of this study. Another variable for future work is the microhardness of composites after aging; others have shown significant effects from this [41,42,43,44].

## 5. Conclusions

Within the limitations of the study, no significant differences in color stability were found between self-blending composite and other composite materials. Accelerated aging and staining had minimal effects on the surface roughness of self-blending composite.

## Figures and Tables

**Figure 1 biomimetics-07-00158-f001:**
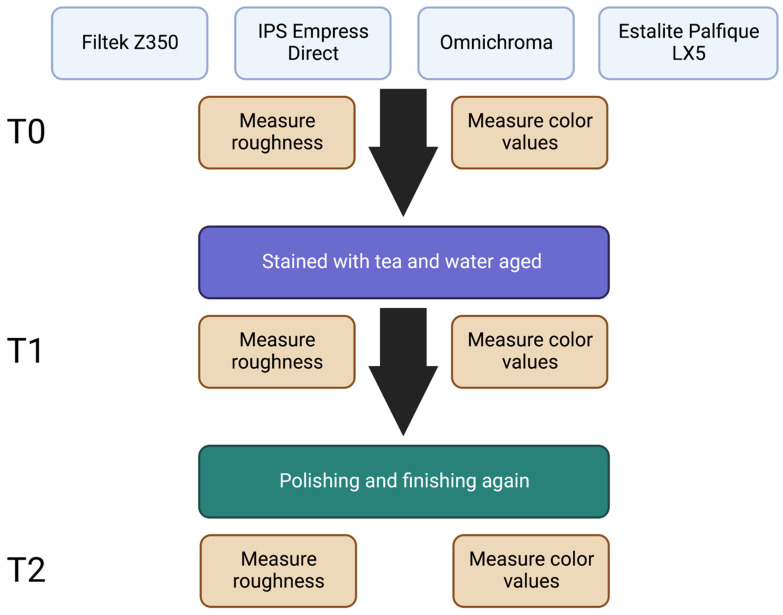
Overall study design. Diagram created with Biorender.com.

**Figure 2 biomimetics-07-00158-f002:**
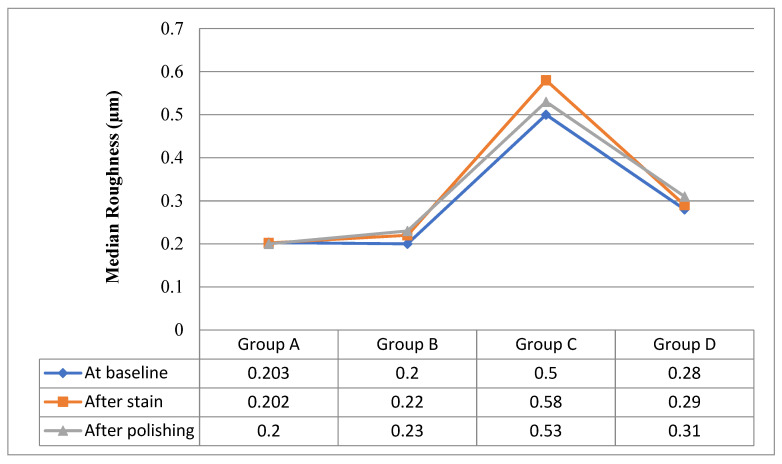
Median composite resin roughnesses at baseline, after staining, and after polishing.

**Table 1 biomimetics-07-00158-t001:** Description of dental composite materials used.

Material	Type	Lot no.	Shade	Filler Weight	Monomer	Manufacturer
Filtek Z350	Nano-filled	NE14538	A2	72.50%	Bis-GMA, UDMA, Bis-EMA, TEGDMA	3M Oral Care St. Paul, MN, USA
IPS Empress Direct	Nanohybrid	Z0199Y	A2	75–79%	Dimethacrylates 20–21.5 wt%	Ivoclar Vivadent Liechtenstein, Zurich
Omnichroma	Supra-nano spherical filler	1081	A1 to D4	79%	UDMA, TEGDMA	Tokuyama Dental, Tokyo, Japan
Estalite Palfique LX5	Supra-nano spherical filler	468	A2	82%	Bis-GMA, Triethylene glycol dimethacrylate.	Tokuyama Dental, Tokyo, Japan

Bis-GMA: Bisphenol-A diglycidlethermethacrylate; UDMA: urethane dimethacrylate; Bis-EMA: Ethoxylatedbisphenol-A dimethacrylate; TEGDMA: triethylene glycol dimethacrylate.

**Table 2 biomimetics-07-00158-t002:** Color-change parameters for resin composites at baseline (T0), after staining (T1), and after polishing (T2).

	Baseline Color (T0) (No = 15)	After Stain (T1) (No = 15)	After Polishing (T2) (No = 15)	Test of Significance (*p*)
**Omnichroma**				(X^2^_Friedman_ = 30, *p* < 0.0001)
**L***	75.8 (74.8–76.0)	69.6 (68.7–70.7)	73.9 (72.3–75.2)	
**Sig detected between episodes**	**{(L1 vs. L0) and (L2 vs. (L0&L1))}**			
**a***	1.6 (1.4–1.7)	4.6 (4.2–4.9)	1.3 (0.8–1.8)	(X^2^_Friedman_ = 27.76, *p* < 0.00001)
**Sig detected between episodes**	**a1 vs. (a0&a2)**			
**b***	13.4 (12.9–13.8)	15.8 (15.1–16.5)	13.1 (11.0–14.0)	(X^2^_Friedman_ = 26.9, *p* < 0.0001)
**Sig detected between episodes**	**b1 vs. (b0&b2)**			
**IPS Empress Direct**				(X^2^_Friedman_ = 28.1, *p* < 0.0001)
**L***	70.0 (69.2–70.5)	63.2 (61.9–70.1)	71.8 (70.2–72.6)	
**Sig detected between episodes**	**L1 vs. (L0&L2)**			
**a***	1.3 (1.1–2.4)	7.6 (6.5–8.1)	1.1 (0.8–1.4)	(X^2^_Friedman_ = 28, *p* < 0.0001)
**Sig detected between episodes**	**a1 vs. (a0&a2)**			
**b***	19.2 (17.3–20.0)	22.5 (18.5–28.0)	18.9 (18.1–19.5)	(X^2^_Friedman_ = 17.4, *p* < 0.0001)
**Sig detected between episodes**	**b1 vs. (b0&b2)**			
**Estalite Palfique LX5**				(X^2^_Friedman_ = 25.2, *p* < 0.0001)
**L***	73.2 (72.1–5.2)	69.4 (68.0–71.0)	74.2 (73.7–75.1)	
**Sig detected between episodes**	**L1 vs. (L0&L2)**			
**a***	0.4 (0.0–1.1)	1.0 (0.9–1.4)	0.7 (0.1–1.1)	(X^2^_Friedman_ = 19.6, *p* <0.0001)
**Sig detected between episodes**	**a1 vs. (a0&a2)**			
**b***	20.5 (19.0–22.4)	22.2 (21.4–23.2)	20.4 (19.1–22.1)	(X^2^_Friedman_ = 17.2, *p* < 0.0001)
**Sig detected between episodes**	**b1 vs. (b0&b2)**			
**Filtek Z350**				(X^2^_Friedman_ = 25.1, *p* < 0.0001)
**L***	75.2 (74.8–75.9)	70.8 (68.3–72.3)	74.7 (74.3–75.3)	
**Sig detected between episodes**	**L1 vs. (L0&L2)**			
**a***	1.2 (1.0–1.4)	1.2 (0.8–1.7)	1.4 (1.2–1.7)	(X^2^_Friedman_ = 20.7, *p* < 0.0001)
**Sig detected between episodes**	**a1 vs. (a0&a2)**			
**b***	12.5 (10.1–14.2)	19.1 (18.0–19.7)	12.0 (11.4–12.9)	(X^2^_Friedman_ = 23.4, *p* < 0.0001)
**Sig detected between episodes**	**b1 vs. (b0&b2)**			

**Table 3 biomimetics-07-00158-t003:** Assessment of color change (after stain vs. after polishing) in each group separately.

	Color Change after Stain (T1)	Color Change after Polishing (T2)	Test of Significance (*p*)
**Omnichroma**	−6.1 (−6.7–0.0)	−1.4 (−2.9–−0.2)	(z= −3.35, *p* = 0.001)
**ΔL***			
**Δa***	3.1 (2.6–3.2)	−0.3 (−0.6–0.1)	(z = 3.42, *p* = 0.001)
**Δb***	2.1 (1.9–3.0)	0.0 (−2.4–1.0)	(z = 3.41, *p* = 0.001)
**ΔE**	7.1 (4.06–7.79)	1.7 (0.22–3.1)	(z = −3.41, *p* = 0.001)
**IPS Empress Direct**			(z = −3.41, *p* = 0.001)
**ΔL***	−6.6 (−8.5–−0.4)	1.8 (−0.3–2.7)	
**Δa***	6.1 (5.0–6.8)	−0.1 (−1.2–0.0)	(z = −3.41, *p* = 0.001)
**Δb***	4.0 (−0.7–9.8)	−0.3 (−1.0–1.8)	(z = −3.23, *p* = 0.001)
**ΔE**	9.54 (7.23–14.21)	1.99 (0.62–2.98)	(z = −3.41, *p* = 0.001)
**Estalite Palfique LX5**			(z = −3.41, *p* = 0.001)
**ΔL***	−3.9 (−7.2–−2.1)	1.3 (−1.2–3.0)	
**Δa***	0.7 (−0.2–1.4)	0.4 (−1.0–1.1)	(z = −3.06, *p* = 0.001)
**Δb***	1.6 (−0.3–3.6)	0.0 (−2.4–2.6)	(z = −3.29, *p* = 0.001)
**ΔE**	4.72 (2.4–7.69)	1.75 (0.24–3.5)	(z = −3.29, *p* = 0.001)
**Filtek Z350**			(z = −3.41, *p* = 0.001)
**ΔL***	−4.8 (−7.0–−3.2)	−0.5 (−1.5–0.2)	
**Δa***	0.0 (−0.4–0.5)	0.3 (0.0–0.7)	(z = 3.2, *p* = 0.001)
**Δb***	6.3 (4.5–9.5)	−0.5 (−2.8–2.6)	(z = 3.4, *p* = 0.001)
**ΔE**	7.75 (6.0–11.4)	1.66 (0.24–3.22)	(z = −3.4, *p* = 0.001)

**Table 4 biomimetics-07-00158-t004:** Comparison between roughness in each group separately for resin composites at baseline (T0), after staining (T1), and after polishing (T2).

Roughness (µm)	Roughness (Baseline)	Roughness (After Stain)	Roughness (After Polishing)	Test of Significance (*p*)
**Omnichroma**	0.203 (0.084–0.543)	0.202 (0.117–0.793)	0.2 (0.1–0.63)	(X^2^_Friedman_ = 4.8, *p* = 0.09)
**IPS Empress Direct**	0.2 (0.09–0.42)	0.22 (0.1–0.57)	0.23 (0.1–0.44)	(X^2^_Friedman_ = 2.8, *p* = 0.247)
**Estalite Palfique LX5**	0.5 (0.27–0.78)	0.58 (0.31–0.9)	0.53 (0.29–0.81)	(X^2^_Friedman_ = 16.8, *p* = 0.0001)
**Sig detected between Roughness episodes**	(Roughness at baseline) vs. (Roughness after stain)			
**Filtek Z350**	0.28 (0.17–0.54)	0.29 (0.22–0.6)	0.31 (0.15–0.51)	(X^2^_Friedman_ = 4.2, *p* = 0.12)

**Table 5 biomimetics-07-00158-t005:** Roughness comparison between groups.

Roughness (µm)	Omnichroma (No = 15)	IPS Empress Direct (No = 15)	Estalite Palfique LX5 (No = 15)	Filtek Z350 (No = 15)	Test of Significance (*p*)
**At baseline**	0.203 (0.084–0.543)	0.2 (0.09–0.42)	0.5 (0.27–0.78)	0.28 (0.17–0.54)	(H= 24.9, *p* = 0.0001)
**Sig between groups**	Group C vs. {group A& group B & group D}				
**After stain**	0.202 (0.117–0.793)	0.22 (0.1–0.57)	0.58 (0.31–0.9)	0.29 (0.22–0.6)	(H= 25.2, *p* = 0.0001)
**Sig between groups**	Group C vs. {group A& group B & group D}				
**After polishing**	0.2 (0.1–0.63)	0.23 (0.1–0.44)	0.53 (0.29–0.81)	0.31 (0.15–0.51)	(H= 27.5, *p* = 0.0001)
**Sig between groups**	Group C vs. {group A& group B & group D}				

## Data Availability

Not applicable.

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
