# Peer review of "Effects of Accelerated Aging on Color Stability and Surface Roughness of a Biomimetic Composite: An In Vitro Study"

_biomimetics, 2022, doi:10.3390/biomimetics7040158_

Round 1
Reviewer 1 Report
The topic of the manuscript is to evaluate and compare colour stability and surface roughness between three different nanofilled composite materials and one self-blending resin composite, based on the in vitro study.
The title, the abstract and the main text of the article are informative. The Introduction briefly presents the issue of nanofilled composites and their properties. The section “Material and Methods” basically explains the chosen study design and should be supplemented. The section “Results” must be improved from the point of view of statistical analysis. The Discussion is interestingly written, however, the clear paragraph about the study limitations is missing. The Conclusions seem to be “take-home” messages.
Some following points must be clarified/corrected for the further processing of this article.
Merits-related comments:
- Please complete keywords with the proper MeSH terms necessary for indexing in the databases.
- In Table 1, please add the LOT numbers of the used composite materials.
- Which lamp was used for the polymerisation (with which fibre diameter?)? Which polymerisation model was used?
- In which tea (black tea, green tea, etc. ) were the samples immersed? Were samples distributed in the solution of the same cold tea for 24 hours?
- During ageing, the samples were immersed in distilled water at what temperature – room temperature or in the incubator that mimics body temperature? Was the water replaced during those 30 days?
- It would be good to add a diagram showing the stages of the study protocol.
- Why did the Authors not carry out the proper post-hoc tests, but only pairwise comparisons?
- What level of significance was determined for the performed statistical analyses?
- It is suggested to add more recent articles from 2020-2022 to the references in the Introduction and Discussion (proposed references: 10.3390/jfb12010001, 10.3390/coatings11060705, 10.3390/polym14153053, 10.3390/polym14102067).
- At the end of the Discussion, the potential study limitations should be clearly explained.
Technical comments:
- Meticulous editorial correction recommended (e. g. “base line” in Figure 1 should be “baseline”, “p<.0001” instead of “p-value<0.0001”).
- The abstract should be a single paragraph and should follow the style of structured abstracts, but without headings.
- Some producers lack information on the city or country of origin.
- “For research articles with several authors, a short paragraph specifying their individual contributions must be provided. The following statements should be used” should be removed.
- References should be described as follows:
1. Author 1, A.B.; Author 2, C.D. Title of the article. Abbreviated Journal Name Year, Volume, page range.
Author Response
Reviewer #1:
Dear reviewer,
We appreciate the evaluation of the manuscript because it enabled us to greatly improve its quality. The following paragraphs provide our point-by-point to each of your comments.
- Please complete keywords with the proper MeSH terms necessary for indexing in the databases.
Response: Recommended change was done.
2 . In Table 1, please add the LOT numbers of the used composite materials.
Response: Recommended change was done.
- Which lamp was used for the polymerisation (with which fibre diameter?)? Which polymerisation model was used?
Response: Thank you for the comment, recommended change was done and information added.
- In which tea (black tea, green tea, etc. ) were the samples immersed? Were samples distributed in the solution of the same cold tea for 24 hours?
Response: Thank you for the comment, clarification was made.
- During ageing, the samples were immersed in distilled water at what temperature – room temperature or in the incubator that mimics body temperature? Was the water replaced during those 30 days?
Response: Thank you for the comment, clarification was made.
- It would be good to add a diagram showing the stages of the study protocol.
Response: Thank you for the comment; we have added a diagram as Figure 1 to describe the flow of the experiment.
- Why did the Authors not carry out the proper post-hoc tests, but only pairwise comparisons?
Response: Thank you for the comment, pairwise comparison was done automatically using SPSS version 25 which is equivalent to post-hoc test. We clarified this in the paper now.
- What level of significance was determined for the performed statistical analyses?
Response: Thank you for the comment, change was made.
- It is suggested to add more recent articles from 2020-2022 to the references in the Introduction and Discussion (proposed references: 10.3390/jfb12010001, 10.3390/coatings11060705, 10.3390/polym14153053, 10.3390/polym14102067).
Response: Thank you for the comment, we added two of these papers about microhardness, which could be a future experiment for us to do.
- At the end of the Discussion, the potential study limitations should be clearly explained.
Response: Thank you for the comment, change was made.
Technical comments:
- Meticulous editorial correction recommended (e. g. “base line” in Figure 1 should be “baseline”, “p<.0001” instead of “p-value<0.0001”).
Response: Thank you for the comment, change was made.
- The abstract should be a single paragraph and should follow the style of structured abstracts, but without headings.
Response: Thank you for the comment, change was made.
- Some producers lack information on the city or country of origin.
Response: Thank you for the comment, change was made.
- “For research articles with several authors, a short paragraph specifying their individual contributions must be provided. The following statements should be used” should be removed.
Response: Thank you for the comment, change was made.
- References should be described as follows:
- Author 1, A.B.; Author 2, C.D. Title of the article. Abbreviated Journal NameYear, Volume, page range.
Response: Thank you for the comment, change was made.
Reviewer 2 Report
This is an interesting comparison of optical properties of different resin composites subjected to staining, accelerated ageing and a new sequence of polishing in a control environment (in vitro study). Materials and methods are fine, and the results are clearly described (perhaps Table 1-bis is not that necessary). The discussion covers all the potential causes for color and roughness changes and is well rooted with backup evidence.
There is a trend for the use of the first person to describe the study, which should be avoided in scientific reports. Please, revise.
Abstract: If T0, T1 and T2 readings were performed with a spectrophotometer and a profilometry, there is no need to confirm that for each reading.
Intro: It is suggested not to use first person in scientific reports. "Our aim in this in vitro study..."
Results: There are two Table 1, one describing the composition of each resin composite and another showing color parameters for L, a and b. My suggestion would be to keep the first one but delete the second, as these results are already described in the plain text.
In Table 2, was the color difference calculated between T0-T1 and T0-T2? or between T0-T1 and T1-T2. Please, clarify this for better understanding of such differences.
Discussion: The heading "Discussion" is missing. Again, try not to use the first person in such a scientific report.
Author Response
REVIEWER #2
Dear reviewer, we thank for your careful reading of the manuscript and your constructive remarks. We have taken the commends on board to improve and clarify the manuscript. Please find below a detailed point-by-point response to all comments.
1. There is a trend for the use of the first person to describe the study, which should be avoided in scientific reports. Please, revise.
Response: Thank you for the comment, change was made.
- Abstract: If T0, T1 and T2 readings were performed with a spectrophotometer and a profilometry, there is no need to confirm that for each reading.
Response: Thank you for the comment, change was made.
- Intro: It is suggested not to use first person in scientific reports. "Our aim in this in vitro study..."
Response: Thank you for the comment, change was made.
- Results: There are two Table 1, one describing the composition of each resin composite and another showing color parameters for L, a and b. My suggestion would be to keep the first one but delete the second, as these results are already described in the plain text.
Response: Thank you for the comment, there is more details in the table and it easier to read. If you do not mind, we would like to keep it to be fully comprehensive and objective in our research report.
- In Table 2, was the color difference calculated between T0-T1 and T0-T2? or between T0-T1 and T1-T2. Please, clarify this for better understanding of such differences.
Response: Thank you for the comment, color difference calculated were made between T0-T1 and T0-T2 as ΔL*, Δa*, Δb*, ΔE at T1 and ΔL*, Δa*, Δb*, ΔE at T2
- Discussion: The heading "Discussion" is missing. Again, try not to use the first person in such a scientific report.
Response: Thank you for the comment, change was made.
Round 2
Reviewer 1 Report
The authors have referred to most of the Reviewers' comments and made significant improvements to the manuscript. However, minor revisions are recommended.
In the statistical methodology, there should be a separate sentence on significance level for all analyses carried out, but in its present form, it looks as if it was only for the Wilcoxon test. Moreover, omitted references on polishing protocols (10.3390/jfb12010001, 10.3390/coatings11060705) are suggested to be added in the Introduction where the Authors cite manufacturers' websites instead of studies (lines 91-94).
Author Response
Reviewer
The authors have referred to most of the Reviewers' comments and made significant improvements to the manuscript. However, minor revisions are recommended.
Answer: We appreciate the positive assessment from the reviewer.
In the statistical methodology, there should be a separate sentence on significance level for all analyses carried out, but in its present form, it looks as if it was only for the Wilcoxon test. Moreover, omitted references on polishing protocols (10.3390/jfb12010001, 10.3390/coatings11060705) are suggested to be added in the Introduction where the Authors cite manufacturers' websites instead of studies (lines 91-94).
Answer: We apologize for the confusion. The level of significance was at 0.05 for all tests and we have modified the sentence to confirm that.
We have replaced the manufacturers’ information with the references as suggested with those two citations and then another two.
